# Improved Stretchable and Sensitive Fe Nanowire-Based Strain Sensor by Optimizing Areal Density of Nanowire Network

**DOI:** 10.3390/molecules27154717

**Published:** 2022-07-23

**Authors:** Rui Li, Xin Gou, Xinyan Li, Hainuo Wang, Haibo Ruan, Yuting Xiong, Xianlun Tang, Yuanyuan Li, Ping-an Yang

**Affiliations:** 1School of Automation, Chongqing University of Posts and Telecommunications, Chongqing 400065, China; lirui@cqupt.edu.cn (R.L.); cqupt_gx@163.com (X.G.); lixinyan2029@163.com (X.L.); whn3542@163.com (H.W.); xiongyt998@163.com (Y.X.); tangxl@cqupt.edu.cn (X.T.); liyy@cqupt.edu.cn (Y.L.); 2China Telecom Sichuan Corporation, Chengdu 610041, China; 3Chongqing Key Laboratory of Materials Surface & Interface Science, Chongqing University of Arts and Sciences, Chongqing 402160, China

**Keywords:** Fe NWs, strain sensor, areal density, PDMS, sandwich structure, sensing mechanism

## Abstract

Flexible strain sensors, when considering high sensitivity and a large strain range, have become a key requirement for current robotic applications. However, it is still a thorny issue to take both factors into consideration at the same time. Here, we report a sandwich-structured strain sensor based on Fe nanowires (Fe NWs) that has a high GF (37–53) while taking into account a large strain range (15–57.5%), low hysteresis (2.45%), stability, and low cost with an areal density of Fe NWs of 4.4 mg/cm^2^. Additionally, the relationship between the contact point of the conductive network, the output resistance, and the areal density of the sensing unit is analyzed. Microscopically, the contact points of the conductive network directly affect the sensor output resistance distribution, thereby affecting the gauge factor (GF) of the sensor. Macroscopically, the areal density and the output resistivity of the strain sensor have the opposite percolation theory, which affects its linearity performance. At the same time, there is a positive correlation between the areal density and the contact point: when the stretching amount is constant, it theoretically shows that the areal density affects the GF. When the areal density reaches this percolation threshold range, the sensing performance is the best. This will lay the foundation for rapid applications in wearable robots.

## 1. Introduction

At present, robotics technology can replace humans to a certain extent to complete certain highly repetitive tasks with high precision requirements. Modern life has put forward higher requirements for service robots with high stretchability, sensitivity (i.e., gauge factor (GF)), fabrication costs, and simplicity [1]. Due to their fragility, most sensors made by using semiconductors or metals will be invalid. Therefore, research into highly stretchable strain sensors that can accommodate the flexible movements of robots, especially limbs and joints, is essential for the development of robotics. Thankfully, a series of flexible strain sensors have been reported with high flexibility, low power consumption, and biocompatibility [2,3].

As a key factor affecting the performance of flexible strain sensors, the sensing units now commonly used in flexible strain sensors are mainly carbon-based material [4,5,6], inorganic oxide nanomaterials [7], and metal nanomaterials [8,9,10]. In these materials, because metal nanowires usually have high electrical conductivity and a high aspect ratio, which can improve tensile strength and creep performance and enhance impact resistance, metal nanowires have been widely used as the sensing unit of flexible tensile strain sensors [11,12]. Currently, Ag nanowires (Ag NWs) [13,14,15] and Cu nanowires (Cu NWs) [16] are the mainstream in research on metal nanowire-based strain sensors. For example, Amjadi M et al. [14] developed sensors based on Ag NWs with high flexibility, stretchability (70%), and GF (2–14). Wang Tao et al. [16] made sensors based on Cu NWs with high flexibility, stretchability (80%), and GF (0.82). However, sensors using Ag NWs and Cu NWs as sensing units still suffer from the inability to achieve high sensitivity and a large strain range at the same time, which limits their application development [17]. The reasons may be that the aspect ratio and conductivity of nanowires are not completely positively correlated with the performance of the sensor, and there is a certain inflection point [9]. Thus, a sensing unit with high conductivity is not necessarily conducive to obtaining sensing performance with a high GF and large strain range. Fe NWs have relatively suitable conductivity, which is expected to overcome the shortcomings of Ag NWs and Cu NWs while obtaining high sensitivity and a large strain range [18]. At the same time, Fe has the characteristics of magnetically controlled construction, which can form a higher aspect ratio by adjusting the magnetic field. The high aspect ratio is conducive to the formation of faster and better conductive paths, which is expected to achieve high sensitivity and a wide range. Further, in addition to the characteristics of metal nanowires themselves, the filling ratio of metal nanowires also has a great impact on the sensing performance, but little attention has been paid to it [13,14]. Studies have explored the mechanism of the synthesis of Fe NWs by the magnetic field-assisted in situ reduction method [19], mastered the method of controlling the aspect ratio of Fe NWs, and studied their potential application in electromagnetic absorption [20]. It was found that the high aspect ratio of Fe NWs is beneficial for obtaining a good conductive network with a low filling mass fraction. This is also conducive to the construction of high-performance sensors, so we further studied the sensing properties of Fe NWs.

The effect of Fe NW areal density on the performance of an Fe NW-based strain sensor from the perspectives of macro and micro mechanisms was studied. In other words, we explored how areal density affects sensor performance and found a way to optimize sensor performance. Experimental results show that the sensor with an areal density of 4.4 mg/cm^2^ has a better sensing performance. The maximum GF value of the sensors is 53, and the corresponding strain range is 15% to 57.5%, with an average nonlinear error of 2.45%. It achieves a balance between relatively high sensitivity and a large strain range, thus laying the foundation for subsequent application in wearable robots.

## 2. Experimental Section

### 2.1. Synthesis and Characterization of Fe NWs

High-aspect-ratio Fe NWs were synthesized according to our previously reported method [19]. First, 2.78 g of FeSO_4_·7H_2_O and 5.30 g of NaBH_4_ were separately dissolved in 100 mL of pure water to prepare solutions. Then, the NaBH_4_ solution was injected into the FeSO_4_·7H_2_O solution at a rate of 10 mL/min. Finally, after the injection, the solution was left to react until no more bubbles appeared, and then a neodymium iron boron (NdFeB) magnet was used to separate and collect the resulting fluffy black precipitate. The products were washed with deionized water and absolute ethanol several times to remove possible residual impurities.

The Fe NW structure was analyzed by XRD (PANalytical, X’pert pro MPD, Malvern, UK) with Cu Kα radiation (l = 0.154 nm, 45 kV at 200 mÅ). The elemental composition and valence of the reactants were determined by XPS (ThermoFisher, Escalab 250xi, Waltham, MA, USA) with Al Kα radiation (hv 1486.6 eV, tv 15 kV). The morphology of the product was observed by SEM (HitachiS-4300) and TEM (JEM 2100F).

Fe NWs had an average aspect ratio of 350 and a diameter of 60 nm, as shown in Figure 1a,b. The XRD patterns (Figure 1c) of the Fe NWs were analyzed. Three distinct distinguishable diffraction peaks were present at 2θ = 44.72°, 2θ = 64.96°, and 2θ = 82.38°, corresponding to (110), (200), and (211) crystals, respectively. In addition, the rings in the SAED pattern further confirmed that the Fe NWs were polycrystals of the bcc structure (Figure 1d).

### 2.2. Fabrication of Fe NW-Based Sandwich Strain Sensor

Studies [21] have shown that PDMS has a very high structural elasticity due to its very low Young’s modulus after curing. However, it does not conduct electricity and cannot be used as a sensor. Fe nanowires were added to provide it with a conductive function. As shown in Figure 2a, the sensor has a typical sandwich structure. It was observed that the conductive layer of Fe NWs was sandwiched between the upper and lower PDMS layers as the encapsulation to effectively prevent the oxidation of Fe NWs [22]. As shown in Figure 2b, in order to fabricate a strain sensor based on Fe NWs, the Fe NW sensing layer was prepared by drawing on the vacuum suction method used by Wang et al. [23]. Firstly, a certain amount of Fe NWs was added to an anhydrous ethanol solution and shaken for 1 min to disperse them evenly in the solution. Secondly, a compound membrane was obtained by using an extraction agent. Thirdly, the composite filter film was cut into a rectangle with a width of 1.6 cm and 0.2 cm. The filter film was removed to obtain the Fe NW conductive film, as shown in Figure 2c,e. After the preparation of the sensing unit, we prepared a flexible substrate and poured the mixed liquid PDMS into a self-made glass groove mold according to a mass ratio of 10:1, as shown in Figure 2d. We precured PDMS at an oven temperature of 70 °C for 6 min to ensure sufficient adhesion [24]. The tailored Fe NW conductive film was adhered to the middle of the precured flexible substrate, and conductive silver paste was applied to both ends of the Fe NW conductive film for further electrical testing. After that, slender flexible copper foil was adhered to the conductive silver paste at both ends to form the electrodes of the flexible tensile strain sensor, as shown in Figure 2f,g. Finally, as shown in Figure 2h, the three-layer PDMS-Fe NWs-PDMS glass groove mold was placed in an oven at 70 °C, cured for 2 h, and then peeled off to form the flexible tensile strain sensor based on Fe NWs.

The constructed Fe NW-based flexible tensile strain sensor was observed through an SEM field-emission electron microscope, as shown in Figure 3. The cross-section refers to the cross-section of the sensor in the direction of 2 cm in length and 1 cm in width, while the longitudinal section refers to the cross-section in the direction of 1 cm in width and uncertain thickness. As shown in Figure 3a, the conducting material Fe NWs were concentrated in a certain area to form a conductive network, which was well wrapped by PDMS to protect it from being oxidized by the external environment. In addition, PDMS itself has a certain degree of flexibility, so it allows the conductive layer to stretch and release with the strain, giving the Fe NW-based flexible tensile strain sensor good flexibility. From the longitudinal section of the sensor (Figure 3b), it was observed that the conductive layer formed by Fe NWs was continuous and complete, which provided a strong electrical guarantee of the tensile sensing capability of the sensor, enabling excellent conductive sensing capability during the strain. Therefore, the flexible strain sensor based on Fe NWs has good flexibility and excellent electrical conductivity.

In this work, we investigated the influence of the Fe NW areal density on the sensing performance. Different Fe NW areal densities were obtained by controlling the amount of Fe NWs dripped during the vacuum filtration step in Figure 2a. Thus, a series of strain sensors with different Fe NW areal densities (areal density varying from 2.2 to 8.8) were fabricated. As shown in Table 1, the areal density is divided into three types: A, B, and C. The areal densities of the three types of samples are 2.2 mg/cm^2^, 4.4 mg/cm^2^, and 8.8 mg/cm^2^, respectively. Additionally, the conductivity of each sample was measured. It can be seen in Table 1 that the higher the surface density, the higher the conductivity. We then further tested the sensing performance of each sample. During the test, the sensor can be affected by external interference, resulting in abnormal results. We eliminated abnormal results and took the average value of the normal test values as the final test result of this category.

### 2.3. Device Construction and Sensing Performance Test

The test system is mainly composed of a Shenzhen Wanshi Electronic Universal Tensile Testing Machine (TestStar ETm102B-TS), a Keithley tabletop multimeter (Keithley2400), and other components, as shown in Figure 4. The electronic universal tensile testing machine was used to measure the applied tensile force. The fixture provided by the electronic universal tensile testing machine was used to fix the sample, and the desktop multimeter was used to measure the initial resistance value of the sensor, with a range of 100MΩ. Before the formal test, it should first be “pre-stretched” to avoid errors in the experimental data due to the relaxation characteristics of the tensile flexible sensor. In the sensing experimental test, the initial extension length of the sensor was 14 mm, the stretching speed of the universal tension machine was set to 0.5 mm/min, and the stretching standard was ASTM D638-08 at a data reading interval time of 0.1 s. The measuring range was 0~100 N, which was recorded on the upper computer in real time.

## 3. Results and Discussion

The sensitivity of the flexible tensile strain sensor is usually described by the GF value, which is represented by the relative change in strain and resistance values, as shown in Formulas (1) and (2). As shown in Figure 5, the resistance of the sensor increases with the increase in strain. The sample with higher surface density has a smaller stretching range, which is because the increase in surface density leads to an increase in conductive layer thickness, thus affecting the stretching range of the sensor.

Interestingly, with the increase in strain, the resistance change trend is divided into two stages (stage I and stage II), and the GF of stage II is larger than that of stage I. Furthermore, it is found that the areal density of the sensing unit has a greater impact on the GF and strain of the sensor. In stage I, the sensor with the larger areal density has a smaller strain range and GF. However, in stage II, it has a larger strain range and GF, while for sample B with an areal density of 4.4 mg/cm^2^, a balance between stage I and II is reached, and the GF and strain obtain better values. Therefore, it can be inferred that this areal density may be the optimal value range for the Fe NW-based strain sensor to achieve better performance. With the further increase in strain, the nonlinear error of the sensor decreases in the second stage compared to in the first stage. At the same time, when the areal density of the sensing unit increases, the nonlinear error becomes smaller. Among them, the optimal value of the areal density is 4.4 mg/cm^2^, the nonlinear error (the average nonlinear error is 2.45%) is the smallest, and it is almost linear.
(1)$$GF=\frac{\frac{∆R}{R_{0}}}{Strain}$$
(2)$$\gamma =\frac{∆Y_{max}}{Y}\times 100\%$$

In summary, for this type of sensor, an areal density of 4.4 mg/cm^2^ (sample B) was expected to achieve the research goals in this paper.

Hysteresis refers to the degree of misalignment between the input curve and the output between the stretching process and the stretching recovery process, as shown in Equation (3). As shown in Figure 6a, this type of sensor exhibits low hysteresis performance during the stretching process and the release process, and the hysteresis error is 8.3%. Among them, when the stretching amount is 25–40%, the hysteresis is very small, and it can be roughly regarded as the coincident state, indicating that the sensor has no hysteresis in this stretch response state.
(3)$$\gamma_{H}=\pm \frac{∆H_{max}}{Y_{FS}}$$

Repeated tests were performed on this type of sensor. It was stretched from 0 to 50% several times, and the current response results were measured. The experimental results are shown in Figure 6b. In the first few iterations of the stretching cycle, the sensor cannot return to the original current value in the next cycle due to the creep of the flexible material. After many cycles, the sensor gradually adapts, and its resistance value tends to change steadily and regularly. The stretching process was repeated many times, and the resistance value returned to the initial value after being stabilized, with good electrical repeatability. When the strain is set to 50%, it can be seen in Figure 6c that this sensor exhibits stable behavior, with monotonic current increasing as the input voltage increases. However, we found that the resistance of the sensor was not stable when the input voltage was between 0 V and 3 V. This was caused by the creep of the flexible material and the hysteresis of the sensor in this strain range. However, with a subsequent input voltage of 3 V to 10 V, it was found that the resistance of the flexible, stretchable strain sensor was in a stable state.

As shown in Figure 6d, the flexible strain sensor was applied with a strain of 69.5% (95% of the total range) and then maintained for a period of time. Over time, the current curve was observed. It was found that as the extension of the sensor increased, the current decreased, and the resistance increased. While maintaining a strain of 69.5%, it was found that the current of the sensor gradually stabilized after a short period of increase. In order to better demonstrate the excellence of the sensor, we compared the performance of different flexible tensile strain sensors based on nanowires, as shown in Table 2. The Fe nanowire-based strain sensor in this paper has a GF of 53 and strain of 57.5%.

## 4. Sensing Mechanism

### 4.1. The Relationship between the Output Resistance and the Contact Point of the Conductive Network

The sensing interior of the sensor was composed of several Fe NWs distributed in PDMS. To further study the sensing mechanism of the flexible tensile strain sensor based on Fe NWs, the Monte Carlo statistical analysis method [27,28] was adopted to analyze the structure of the conductive network formed by sensitive elements in the sensor. Figure 7a shows the case of one unit included in conductive networks formed by interlaced Fe NWs. When the unit was stretched to ∆*L*, the conductive network of the Fe NWs of this unit became sparse, making its contact points in the original unit area greatly reduced.

As shown in Figure 7b, the Fe NWs inside the sensor exist in a three-dimensional space in PDMS. To analyze the variable output resistance of the sensor, electrical analysis of the Fe NWs is required: the resistance of the nanowire forms a bridge network [23]. The contact points between the nanowires are shown in Equation (4). The internal resistance of each nanowire sensor can be considered substantially equal, and as shown in Equation (5), the output resistance can be obtained, where *R_Adjacent total_* represents the resistance of each Fe NW, *P_contact_* represents the contact point between the nanowires inside the sensor, *i* represents the number of *i* nanowires connected (including contacts) in the stretched state, and *N* represents the total number of nanowires in the sensor.
(4)$$P_{contact}=\frac{i\left(i-1\right)}{2}\;\left(1<i<N\right)$$
(5)$$R_{Output}=P_{contact}R_{Adjacent\;total\;}\;\left(1<i<N\right)$$

By simplifying the actual three-dimensional relationship of the nanowires in a certain direction, we found that there are three relationships between two nanowires, as shown in Figure 7c, including connection without contact resistance, connection with contact resistance, and separation resistance ∞. In the following equations, *d* represents the distance between two nanowires in the same plane, and *d_c_* represents the maximum distance when there is contact resistance between two nanowires. Thus, the output resistance of the sensor can be expressed as shown in Equation (6). According to the electron tunneling theory, the contact resistance of the resistance can be obtained, as shown in Equation (7). Equation (7) reflects the relationship between increasing contact points and increasing output resistance as the amount of stretching increases, which was also verified in experiments. At the same time, the reason why the output resistance of the sensor has different growth stages, that is, why the GF differs, is found as the amount of stretching increases. When the stretching has just begun, the contact points gradually increase, but there is no contact resistance inside the sensor, and the output resistance mainly increases according to a quadratic function. As the amount of stretching increases to a certain value, contact resistance appears inside the sensor, and the output resistance value is expressed by a quadratic function growth distribution and a normal distribution; the growth trend accelerates, and two stages (I and II) appear. Further, Equation (8) is obtained, and the output resistance has a necessary relationship with the contact point of the conductive network formed by the nanowire. When the nanowire connection in the sensor has no contact resistance, the output resistance will exhibit quadratic function growth. When the nanowire connections all have contact resistance, the output resistance presents a normal distribution trend. When the nanowires are not connected, the output resistance is infinite.
(6)$$R_{Output\;\;}\left(P_{contact}\right)=\left\{\begin{array}{c} 2P_{contact}R_{\;}\;\left(1<i<\frac{N\left(N-1\right)}{2},d=0\right)\\ P_{contact}(2R+R_{tunnel}{}_{\;})\;\left(1<i<\frac{N\left(N-1\right)}{2},0<d<d_{c}\right)\\ \infty \;\;\;\;\;\left(\frac{N\left(N-1\right)}{2}<i<N,d>d_{c}\right) \end{array}\right.$$
(7)R_tunnel=h^2 d/Ae^2 √2mφ  exp4πd/h √2mφ
(8)$$F\left(P_{contact}\right)=\left\{\begin{array}{c} \int_{1}^{\frac{N\left(N-1\right)}{2}}2Rd{\left(P_{contact}\right)}_{\;}\;\;\;\\ \int_{1}^{\frac{N\left(N-1\right)}{2}}\int_{0}^{d_{c}}(2R+\frac{h^{2}d}{{\mathrm{Ae}}^{2}\sqrt{2m\varphi }}\exp(\frac{4\pi d}{h}\sqrt{2m\varphi }))\\ \int_{\frac{N\left(N-1\right)}{2}}^{N}\int_{d_{c}}^{\infty }\infty d\left(P_{contact}\right) \end{array}\right.d(P_{contact})$$

### 4.2. The Relationship between the Output Resistance and Areal Density

The areal density of the sensor is shown in Equation (9), where m_0_ represents the mass of a single nanowire, *L*_0_ represents the original length of the sensor, ∆*L* is the varying length of the sensor during the stretching process, and $a_{width}$ is the width of the sensor. It is further found that there is an opposite theoretical seepage relationship between Fe NW density and sensor resistance, as shown in Figure 8.

As the tensile force increases, the areal density of Fe NWs in the sensor gradually decreases, the contact points of the conductive network gradually become sparse, and the resistivity of the sensor slowly rises. Until the areal density of the Fe NWs reaches a critical value (the percolation threshold [24]), the resistivity of the sensor rises sharply. At this time, the sensor quickly completes the transition from conductor to semiconductor or even insulator, and its resistivity increases by several orders of magnitude, as shown in Equation (10).
(9)$$\phi_{Area\;density}=\frac{N\cdot m_{0}}{S}=\frac{N\cdot m_{0}}{\left(∆L+L_{0}\right)a_{width}}$$
(10)$$\rho_{Output\;}=\rho_{Fe\;NWs}{\left(\phi_{Area\;density}-\phi_{Percolation\;threshold}\right)}^{-t}$$
(11)$$R_{Output\;}=\rho_{Output}\frac{\left(∆L+L_{0}\right)}{\left(∆L+L_{0}\right)a_{width}}$$
where t is the structure constant of the sensor network, which is inversely proportional to the current areal density under the condition of constant stretching [29,30]. Equation (12) of sensor output resistance can be obtained from Equations (10) and (11):(12)$$R_{Output\;}=\rho_{Fe\;NWs}{\left(\phi_{Area\;density}-\phi_{Percolation\;threshold}\right)}^{-\frac{A}{\phi_{Area\;density}}}\frac{\left(∆L+L_{0}\right)}{\left(∆L+L_{0}\right)a_{width}}$$
where A is a constant, and due to the existence of t, the areal density of sensors with different gradients may face different trends of resistivity growth. When the amount of stretching is fixed, the output resistance of sensors with different areal densities is proportional to the resistivity. However, how do we achieve good sensitivity and linear response curves? This is what we explored It was verified in the experiment that the density of sample B is closest to this percolation threshold, and the nonlinear error is the smallest, which can make the output resistance value increase linearly. When the density is 4.4 mg/cm^2^, the sensor performance is the best, which can indicate that the optimal surface density is near this density. Equation (13) can also be obtained from Equation (9):(13)$$N=\frac{\phi_{Area\;density\left(∆L+L_{0}\right)a_{width}}}{m_{0}}$$

From this, the relationship between the contact points of the conductive network and the areal density can be obtained, as shown in Equation (14). When the stretching amount is fixed, the contact points of the conductive network inside the sensor are positively correlated with the areal density of the sensor. The greater the density of the sensor, the more contact points.
(14)$$P_{contact}=\frac{i\left(i-1\right)}{2}\;(1<i<\frac{\phi_{Surface\;density\left(∆L+L_{0}\right)a_{width}}}{m_{0}})$$

## 5. Conclusions

In this work, we found that the sensing performance of the sensor (expressed as the GF and strain range) is related to the areal density of Fe NWs in the sensing unit. The sensor model was obtained by analyzing the relationship between the output resistance, the contact point of the conductive network, and the density of the sensor unit. In addition, it was found that the areal density and the sensing performance do not have an absolute positive correlation, but there is a percolation threshold. When the areal density reaches this percolation threshold range, the sensing performance is the best. Experiments showed that area density of 4.4 mg/cm^2^ has excellent sensing performance and is closest to the percolation threshold. Tunable gauge factors of the sensors are in the ranges of 37 to 53, the corresponding strain range is 15% to 57.5%, and the average nonlinear error is 2.45%. Thus, we developed a new sandwich-structured strain sensor based on sandwich-structured Fe NWs that has a high GF with a large strain range and low hysteresis, and the sensor is stable and low-cost. In future work, we will extend our work to explore other influencing factors (e.g., temperature and sensing structure) on the performance of Fe NW-based strain sensors to achieve both a higher GF and large strain range for quickly applying to wearable robots.

## Figures and Tables

**Figure 1 molecules-27-04717-f001:**
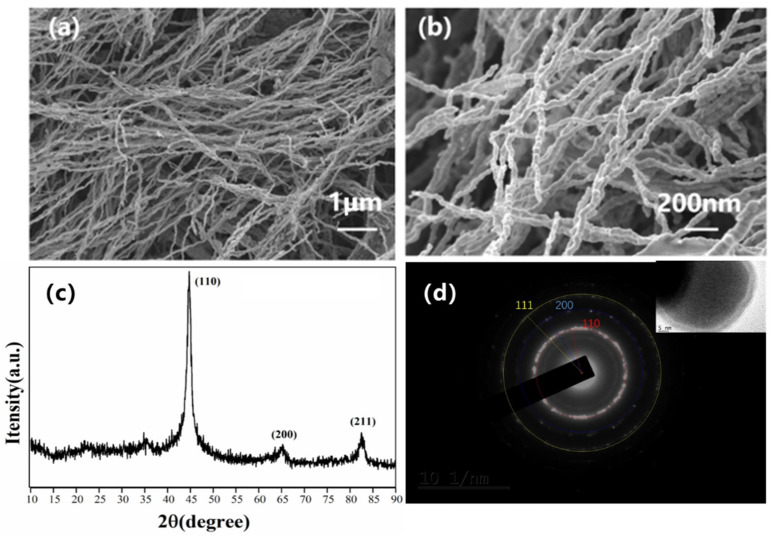
Characterization of Fe NWs: (**a**,**b**) remote and local SEM of Fe NWs, (**c**) XRD of Fe NWs, and (**d**) SAED of Fe NWs.

**Figure 2 molecules-27-04717-f002:**
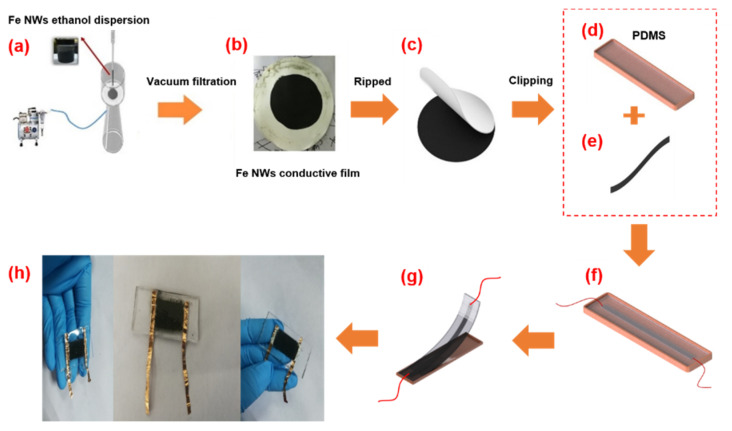
Construction process of flexible sensor based on Fe NWs: (**a**) vacuum filtration, (**b**) transfer, (**c**) shaping, (**d**) PDMS substrate, (**e**) tailoring, (**f**) sensor assembly, (**g**) sensor molding, and (**h**) an image of the sensor.

**Figure 3 molecules-27-04717-f003:**
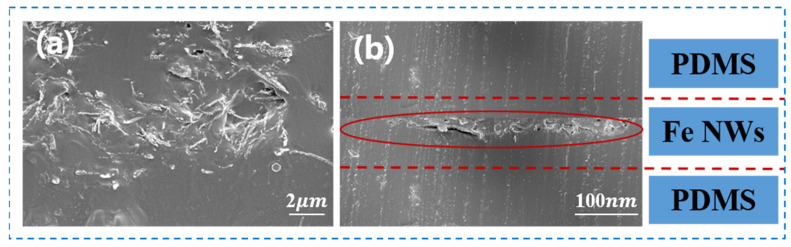
Characterization of Fe NW-based sandwich strain sensor. (**a**) Cross-section; (**b**) longitudinal section.

**Figure 4 molecules-27-04717-f004:**
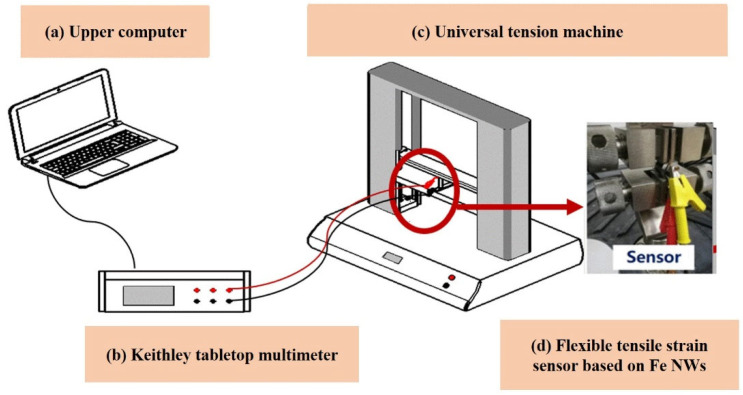
Experimental construction and mechanical performance analysis of a flexible tensile strain sensor based on Fe NWs: (**a**) display device, (**b**) collection device, (**c**) strain tensile test, and (**d**) flexible tensile strain sensor based on Fe NWs.

**Figure 5 molecules-27-04717-f005:**
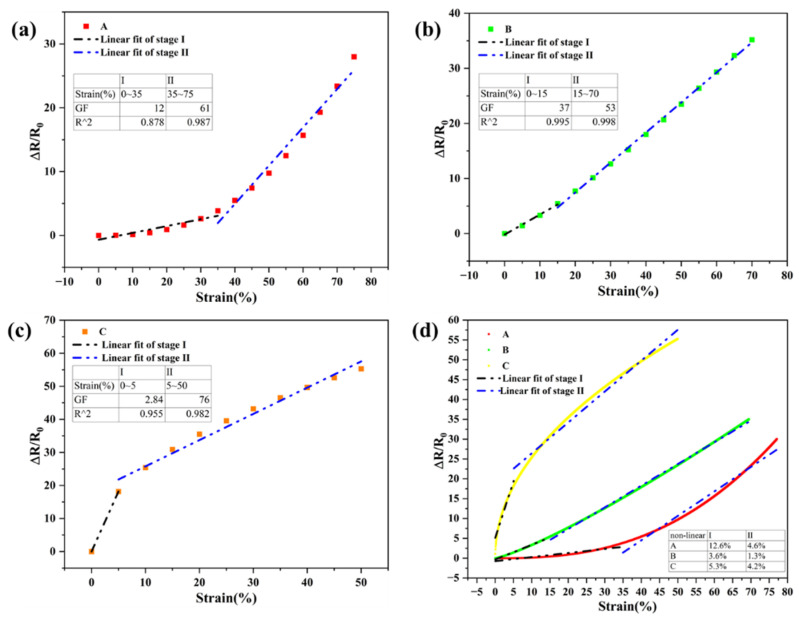
The GF and working range of Fe NW-based flexible tensile strain sensor: (**a**) the GF of sample A changes with the increase in stretching, (**b**) the GF of sample B changes with the increase in stretching, (**c**) the GF of sample C changes with the increase in stretching, and (**d**) the nonlinear error of samples A, B, and C.

**Figure 6 molecules-27-04717-f006:**
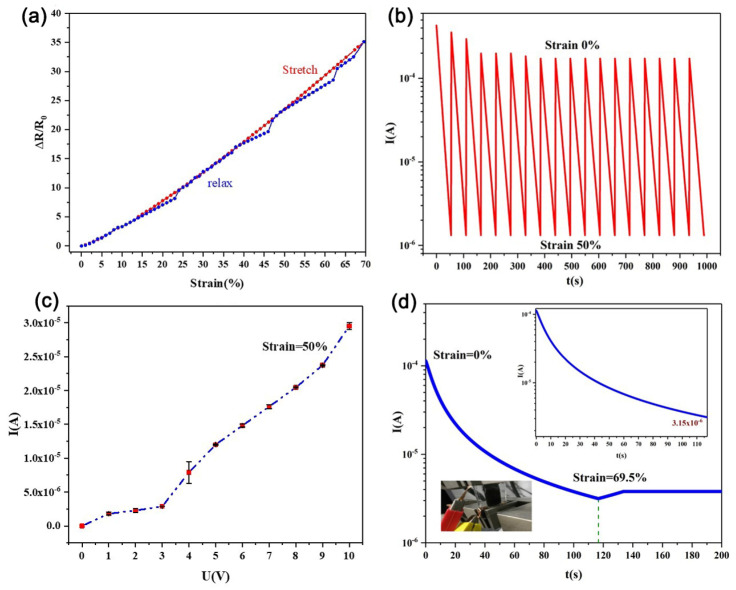
Sensor performance of Fe NW-based flexible tensile strain sensor: (**a**) the hysteresis performance of the sensor when it is stretched to 70%, (**b**) the repetitive performance of the sensor when it is repeatedly stretched from 0 to 50%, (**c**) when the sensor is stretched to 50%, as the voltage increases, the sensor output current changes, and (**d**) sensor output current response.

**Figure 7 molecules-27-04717-f007:**
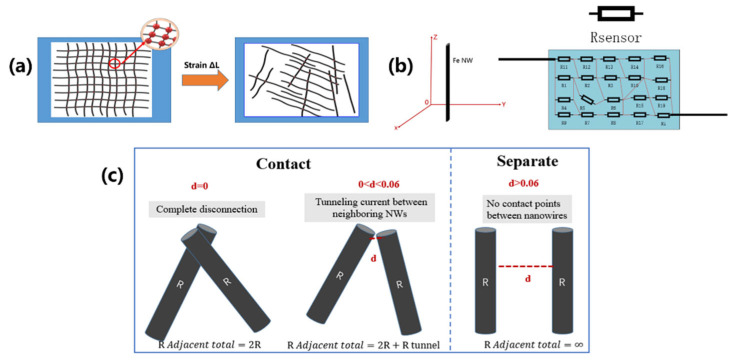
Analysis of the microscopic mechanism of the sensor: (**a**) with stretching, the contact points of the nanowire network are distributed, (**b**) sensor output resistance state model, and (**c**) the relationship between the nanowires inside the sensor.

**Figure 8 molecules-27-04717-f008:**
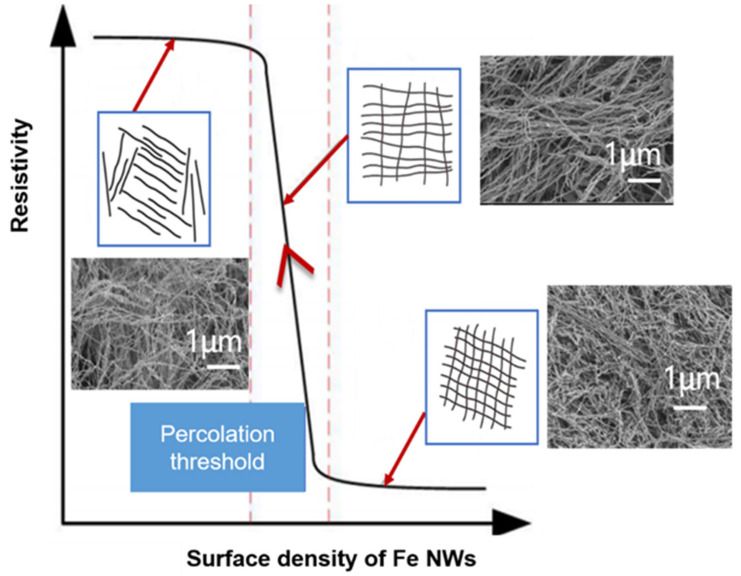
Percolation theory of flexible tensile strain sensors based on Fe NWs.

**Table 1 molecules-27-04717-t001:** Sensors based on Fe NWs with different areal densities.

Sample Sensor	Areal Density (mg/cm^2^)	Conductivity (S/cm)
Class A	2.2	$1.4\times 10^{-4}$
Class B	4.4	$1.8\times 10^{-3}$
Class C	8.8	$7.3\times 10^{-3}$

**Table 2 molecules-27-04717-t002:** Key performance of flexible tensile strain sensor based on nanowires.

Class	GF	Strain
Au NWS [10]	1.82~7.38	14–25%
Ag NWS [14]	2~14	70%
Cu NWS [22]	0.82	80%
CNT [25]	1140	8.75%
Graphene [26]	>1000	5%
Fe NWs	53	57.5%

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
