# Peer review of "Improved Stretchable and Sensitive Fe Nanowire-Based Strain Sensor by Optimizing Areal Density of Nanowire Network"

_molecules, 2022, doi:10.3390/molecules27154717_

Round 1
Reviewer 1 Report
The authors have addressed all my previous comments. So the manuscript can be accepted in the current form.
Author Response
Reply to reviewer #1

Reviewer 2 Report
A sandwich structure strain sensor based on Fe NWs is proposed and experimentally investigated in this work. It is very interesting and novel job. The strain sensor possess good performances, for example, it has a higher GF (37-53) in a larger strain range (15%-57.5%), low hysteresis (2.45%), stability and low-cost sensors with the density of Fe NWs in 4.4 mg/cm2. Additionally, the effect of Fe NWs areal density on the performance of Fe NW-based strain sensor from the macro and micro mechanisms is discussed. The paper is well organized and the results support the conclusions. I think it can be accepted.
1. The language should be improved again, some grammatical errors exist.
Author Response
Reply to reviewer #2

Reviewer 3 Report
In this paper, the author proposed a Fe NW-based strain sensor, including the effect of the volume fraction of Fe NWs particles. There are some issues for possible publication in this journal. There are some comments listed below:
1. It isn't obvious for readers to understand the experiment steps. So please insert the images of Fe NWs conductive film.
2. Please add the additional dynamic characteristics like the cycling strain test or frequency test to better understand the proposed sensor performance
3. Please insert an experiment of temporal variation in resistance when applied strains change, for example, 25%, 50%.
4. Please insert data that shows the relative change in the resistor and corresponding stress of the sensor at a specific strain point.
5. In figure 8, please point out the volume fraction of conductive particles and estimate the area of the percolation transition range.
6. The author should add the conductivity data to show the dependence of the percolation threshold with the volume fraction of Fe NWs.
7. In section 4.2, please point out more clearly why an areal density of 4.4 mg/cm2 is the best.
8. Please, suggest any possible application using the proposed sensor.
Author Response
Reply to reviewer #3

Round 2
Reviewer 3 Report
I agree with the revision of the manuscript, so I recommend it for publication.
This manuscript is a resubmission of an earlier submission. The following is a list of the peer review reports and author responses from that submission.
Round 1
Reviewer 1 Report
In the submitted manuscripts authors investigated the effect of different Fe nanowires density (2.2 mg/cm2, 4.4. mg/cm2 and 8.8 mg/cm2) on the strain value of the flexible PDMS piece by measuring conductance.
Here you can find the questions and comments:
Q1. As seen from the synthesis method and Fig.3, the Fe nanowires filled a volume not an area on the surface. The nanowires dispersed randomly and not homogenous in the displayed area in Fig.3. Why authors evaluated it as an areal density?
Q2. In the synthesis section, what is the appropriate amount of FeSO4.7H2O? What is certain rate for injection of NaBH4? Authors should use concrete experimental data.
Q3. In figure 1, the figures are not in same dimension and properly presented. What is the meaning of JCPDS:Card NO.65-4899 inset information in Figure 1c? It causes misunderstanding. Authors should remove it, or they should prepare a comparison graph by putting both literature XRD-data (by citing) and experimental data.
Q4. In figure 2, the process flow is not clear. The process is going on from c to e or d?
Q5. In the lines from 139 to 141, the authors mentioned about preparation of 6 batches for each density values. And they mentioned that they removed some of the data and took the average value of the normal test values. First, what is the abnormal result? How many abnormal results you get from each batch? Second, why you removed? Third, how many data you used for determination of the average value?
Q6. Did author prepared an empty PDMS sensor to reveal the effect of nanowires?
Author Response
Reply to reviewer 1

Reviewer 2 Report
The present manuscript entitled “Improved Stretchable and Sensitive of Fe Nanowire-Based Strain Sensor by optimizing areal density of nanowire network” by Rui Li et al., describes the sensor sensing performance (such as GF, strain range) is related to the areal density of Fe Nanowires in the sensitive unit. By analyzing the relationship between the contact point of the output resistance and the conductive network, and the density of the sensing unit, the sensor model is attained. From the results, it is obtained that the sensor performance with an areal density of 4.4 mg/cm2 is the best. The authors report an interesting approach but the presentation of the work is clear. The objective and justification of the work are clear, and the experimental work is significant. The study is accurate and adequate, and thus, I would recommend it for publication in Molecules. However, certain Minor issues are detailed below to improve the quality of the manuscript.
I advise the authors to take the following points into account while revising their manuscript.
Comment 1: There are some typographical errors in the manuscript text, so the authors need to correct them in the revised manuscript. In the whole manuscript, the authors must be taken care of the superscripts and subscripts, and abbreviations. For e.g. Line 24 and 68, 4.4 mg/cm2 should be 4.4 mg/cm2.
Comment 2: Abstract is poorly written, it needs to be revised.
Comment 3: The authors need to add and discuss some more recent literature in the introduction section to strengthen the background of their work.
Comment 4: The authors should explore and discuss the SEM and XRD results (Section 2.1.) with some more references to prepare a better discussion.
Comment 5: In figures 3(a) and 8, the scale bar is written um should be µm
Comment 6: The homogeneity of the reference section needs to be maintained. Some references are abbreviated, and some are not. Please check and revise accordingly to the journal's instructions: https://www.mdpi.com/journal/molecules/instructions.
Author Response
Reply to reviewer 2

Round 2
Reviewer 1 Report
Dear Authors,
I checked the corrections in the manuscript entitled “Improved Stretchable and Sensitive of Fe Nanowire-Based Strain Sensor by optimizing areal density of nanowire network”. Thanks for the corrections, I put my comments on the revised manuscript below;
1. As seen from the synthesis method and Fig.3, the Fe nanowires filled a volume not an area on the surface. The nanowires dispersed randomly and not homogenous in the displayed area in Fig.3. Why authors evaluated it as an areal density?
Response: Thanks to the reviewer for this comment. The different Fe nanowires area density were calculated by controlling the amount of Fe nanowires dripped during the vacuum filtration step in Figure R1(a). Thus, Fe nanowires density was used to define different samples. (We have added these to the lines from 152 to 153 in the manuscript.)
Comment 1) Thanks to author for their answer however I was wondering why you have not calculated volumetric density of the nanowires? Fe nanowires are filling an 3D PDMS substrate. Is thickness of PDMS substrate not effective on the conductance?
2. In the synthesis section, what is the appropriate amount of FeSO4•7H2O? What is certain rate for injection of NaBH4? Authors should use concrete experimental data.
Response: Thanks for the reviewer’s careful review and suggestion. It is very important to specify the appropriate amount and certain rate for injection, but we ignored it. The appropriate amount of FeSO4•7H2O and NaBH4 is 2.78 g and 5.30 g, respectively. They were dissolved in pure water to make 0.1mol/L and 1.4mol/L solutions. And the rate for injection of NaBH4 is 10 mL/min. (We have added these to the lines from 78 to 80 in the manuscript.)
Comment 2) Thanks for revision but I strongly recommend the use of numerical values instead of appropriate amount, certain rate terms. Please remove these ambiguous terms from the sentences and put numerical values that you used in the experiment.
5. In the lines from 139 to 141, the authors mentioned about preparation of 6 batches for each density values. And they mentioned that they removed some of the data and took the average value of the normal test values. First, what is the abnormal result? How many abnormal results you get from each batch? Second, why you removed? Third, how many data you used for determination of the average value?
Response: Thanks for the reviewer’s careful review. During the performance test, the sensor will be affected by external interference, resulting in the abnormal result. Therefore, we adopt 3 criteria to detect and exclude outliers. The same test was performed for all samples, but the number of outliers was variable. The change curves of the same type of sensor are averaged to obtain the final curve in Figure R3. (We have added these to the lines from 158 to 159 in the manuscript.)
Comment 5) What is 3 criteria? I can understand experimental errors while preparing the sensors which is quite possible for a hands-on work with flexible materials. However, reporting that you ignored the abnormal data looks quite problematic. It refers that your sensors are not reliable. You indicated in your answer that “The same test was performed for all samples” however in Figure R3. The data scales of strain values are different. For example, you have not measured more than 50% strain value for sample C. Similarly, strain value scale is different for Sample A and B.
6. Did author prepared an empty PDMS sensor to reveal the effect of nanowires?
Response: Thanks for the reviewer’s careful review and suggestion. We have made a pure PDMS sensor, which the resistance of it is too large to be detected by the instrument. Studies[1] have shown that PDMS has a very high structural elasticity due to its very low Young's modulus after curing. However, it does not conduct electricity and cannot be used as a sensor. Fe nanowires are added to make it have conductive function.
Comment 6) It could be better to add the reference to the suitable place by mentioning why there is no measurement with pure PDMS sensor.
Comment 7) I checked Ref. 20. (line 81-82) It was not reported by any authors and their institutions in the submitted manuscript. This is quite problematic.
Comment 8) In chapter 4.1, you have not given enough reference for sensing mechanism.
Comment 9) Line 266-270, there are so many repetitions.
Comment 10) In line 288, Ref 26 is not related with the context. What is the relationship between percolation threshold and referred study?